# Nurse’s Evaluation on Health Education in Portuguese Pediatric Hospitals and Primary Care for Children/Young and Parents

**DOI:** 10.3390/children9040486

**Published:** 2022-04-01

**Authors:** Anabela Fonseca Pereira, Joaquim Escola, Vitor Rodrigues, Carlos Almeida

**Affiliations:** 1Higher School of Health, Portuguese Red Cross Alto Tâmega, 5400-673 Chaves, Portugal; anabelapereira83@gmail.com; 2Institute of Philosophy of the University of Porto, School of Human and Social Sciences, University of Trás-os-Montes e Alto Douro, 5000-801 Vila Real, Portugal; jescola@utad.pt; 3Research Center in Sports Sciences, Health Sciences and Human Development, CIDESD, 5000-801 Vila Real, Portugal; vmcpr@utad.pt; 4School of Health, University of Trás-os-Montes e Alto Douro, 5000-801 Vila Real, Portugal

**Keywords:** health promotion, pediatrics, nursing care

## Abstract

Aim: This study aimed to analyze the nurse’s evaluation of the health education practice to children and parents. Methods: This is a descriptive and transversal research with a quantitative approach. The selection took place by non-probabilistic convenience sampling, and was developed with nurses on health units for pediatric hospitalization and primary health care in northern Portugal (Trás-os-Montes and Alto Douro Hospital Center, EPE, Northern Regional Health Administration, Northern Local Health Unit, EPE.). Data were collected using a questionnaire with a sample of 311 nurses in the second semester of 2018. Results: 77.5% (*n* = 241) of nurses perform health education daily; 65% (*n* = 202) prepare according to the identified needs; the “Identification of children/young and parents’ health priorities” was considered to be the most facilitating element (*n* = 279; 89.7%); the most difficult element was the “Fear of confidentiality breach by the children/young and parents” (74.6%; *n* = 232); and 65.9% (*n* = 205) of nurses considered this practice to have equal importance compared to other nursing interventions. It was also found that academic/professional qualifications and the place of professional practice influence the importance that nurses attribute to HE practice. Conclusions: We can state that there is an appreciation of the binomial child and parents for a better identification of needs, and of the importance attributed to the current legal guidelines (letting themes leading to good health practices be addressed by nurses), which translates into a practice capable of influencing the determinants of health, which promotes health-enhancing behaviors and thus both leads to health gains and reinforces the nurse’s position as health-promoting agents.

## 1. Introduction

The discussion about child health care has been going on since the First International Conference on Health Promotion, which took place in 1986 and is described in the Ottawa Charter [1]. Since then, childcare has been looked at as a priority on the global governmental organization’s agendas. Parenting support and providing skill sets to parents is necessary in order to obtain a support structure to improve parenting skills [2]. In addition, “The Royal College of Pediatrics and Child Health”, at its annual meetings, is also dedicated to research on global child health, and universities in Italy, the Netherlands, Norway and Sweden have institutes dedicated to international maternal and child health [3].

In the Ottawa Charter, the importance of the health promotion (HP) concept was also mentioned as being a “process of empowering the community to act to improve the quality of life and health, including a greater participation in the control of this process” [1]. Since children’s health is an important requirement for the society’s development, the promotion of child health requires health professionals to develop their skills in order to perform effective parental interventions [4].

Health education (HE) assumes itself as not only an educational practice that promotes health and prevents diseases, but also as a theoretical learning process that integrates different types of knowledge (scientific, popular and common sense), thus enabling both the acquisition of new habits and the decision making in health by individuals involved [5]. HE is therefore a set of opportunities consciously built for learning, aimed at increasing health literacy [6].

### Health Education in Portugal

In the last decade, the Portuguese health status has improved considerably. However, health inequalities are linked to a series of determinants of health (stress, people’s conditions, physical environment) and behavioral risk factors (tobacco, alcohol, diet and physical inactivity). It should be noted that, contrary to the normal trend, there is a sharp decline in smoking, especially among 15-year-olds, which is a positive result and the outcome of public health actions aimed at tobacco control. Infant mortality has declined but injuries and trauma are still the number one cause of death. The rate of physical inactivity among 15-year-olds is among the highest in the European Union, and it is estimated that 10 to 20% of children will suffer from some mental health problem.

Regarding obesity, its prevalence in 15-year-olds remains close to the European Union (EU) average, but the rates of physical inactivity are among the highest in EU countries, so national nutrition strategies have been implemented, as well as the prevention and treatment of obesity and promotion of physical activity [7].

For children up to 6 years of age, there are programs with intersectoral responsibility and interdisciplinary approaches [8]. There are also programs with the objective of promoting health decision making, self-confidence and the mental well-being of vulnerable pre-adolescents [9].

In a study on health literacy levels in 2015, with recourse to the “European Health Literacy Survey” (used in a European study), it was found that 11% of Portugal’s population had an “inadequate” level of literacy, 38% had a “problematic” level of literacy and only 50% had a “sufficient” level of literacy. Regarding health care literacy, 45.4% had limited literacy (this is once again below the average values of other participating countries) [10]. In this scenario, the HE practice stands out as a valuable tool, as nurses play an important role in empowering the individuals in their health promotion [11]. Thus, both the role of nurses as educators in the health promotion process [2] and their technical and human capacity to meet the individuals and family’s needs [12] are recognized. This research study has the objective of identifying the nurse’s evaluation on the HE practice to children/young and parents.

## 2. Methods

This is a descriptive and transversal study with a quantitative approach, regarding the nurse’s evaluation of the HE practices in children/young and parents, and is part of an investigation on the contribution of the HE practices provided by nurses to the health of the children/young and parents.

This research was developed in two hospitals and in primary health care facilities at Trás-os-Montes and Alto Douro province, in a total of 4 pediatric hospitalization services and 29 health centers—Trás-os-Montes and Alto Douro Hospital Center, EPE, Northern Regional Health Administration, Northern Local Health Unit, EPE, during the second semester of 2018. The option for a non-probabilistic sampling of convenience and geographical area was due to the advantage of having better accessibility to the population.

The inclusion criteria for the target population were: being a nurse for more than 6 months in a pediatric hospital inpatient ward; being a nurse for more than 6 months in primary health care (whose activity involves attending children health appointments); and nurse’s acceptance to participate in the study. The sample consisted of 311 nurses (78.5% of the study population).

In the data collection, a questionnaire survey was built and used, which was applied after authorization by those responsible at the institutions that participated in the study and a favorable opinion from the respective ethics commissions (Northern Regional Health Administration: authorization n. 124/2018; Trás-os-Montes and Alto Douro Hospital Center, EPE: authorization n. 256/2018; Northern Local Health Unit, EPE: authorization n. 00316/2018). After applying 20 interviews to parents, the questionnaire was constructed, with three sections: sample characterization, HE practices performed by nurses (5 questions) and a scale that measured HE of children/young and parents (EAEPS), which contained 48 items and was validated [13]. The recommended ethical procedures were also followed. There was no relationship between the researcher and the study participants, which decreased the researcher’s potential to influence the study or for the study to influence the researcher. It was also ensured that the distributed questionnaires included the informed consent, explaining to the participants that their participation would not entail risks, and that they had the full freedom to accept or leave the study without restrictions or consequences. They were given a guarantee of confidentiality regarding the preservation of the collected data, which would be used exclusively for research. Privacy was also respected and the anonymity of the participants was ensured.

The collected data were analyzed using the statistical program *Statistical Package for the Social Sciences* (IBM Corp. Released 2013. IBM SPSS Statistics for Windows, Version 22.0. Armonk, NY: IBM Corp.). To analyze the sociodemographic characterization, a descriptive analysis was used (frequency distributions, measures of central tendency and measures of dispersion). For associations between variables, a statistical inference was used [13]. The Kolmogorov–Smirnov test revealed that the sample did not present a normal distribution, so the non-parametric chi-square (χ) test [14] by Monte Carlo simulation was used. For all statistical tests, a significance level of 0.05 (*p* < 0.05) was adopted.

## 3. Results

### 3.1. Sociodemographic and Professional Characterization

The results showed that, for the total number of valid cases (*n* = 311), 89.1% (*n* = 277) were female and 10.9% (*n* = 34) were male; the age range was between 40 to 50 years; 35.4% (*n* = 110) belonged to the “Nurse” category, as well as the “Specialist nurse” category (35.4%), and 29.3% (*n* = 201) to the “Graduated nurse” category; 56.6% (*n* = 176) only had a “Graduation” and 39% (*n* = 121) had a “Post-Graduation of Specialization”; the time of professional practice was in the range of 15 to 20 years; 86.5% (*n* = 269) performed their professional activity in primary health care and 13.5% (*n* = 42) in hospital.

### 3.2. HE Practices Performed by Nurses

The results showed that 77.5% (*n* = 241) carried out HE daily, 13.5% (*n* = 42) carried out HE weekly, 8% (*n* = 25) carried out HE occasional and 1% (*n* = 3) answered never.

It was found that 65% (*n* = 202) of the respondents admitted to preparing the HE practice according to the identified needs, 13.2% (*n* = 41) stated that the planning was based on improvisation, 10.6% (*n* = 33) reported carrying out the preparation according to elaborated guidelines/norms and 11.3% (*n* = 35) reported carrying it out based on all options.

The most important topics to be approached during HE practice were: “Healthy eating” (61.4%; *n* = 191); “Accidents prevention” (57.6%; *n* = 179); “National Vaccination Plan” (50.2%; *n* = 156) and “Harmful behaviors prevention” (45%, *n* = 146).

Nurses also considered that the identification of children/young and parents’ health priorities was the most facilitating element of this practice (*n* = 279; 89.7%), and “the fear of confidentiality breach by children/young and parents” (74.6%; *n* = 232) was considered to be the most difficult element.

When asked about HE practice importance, compared to other nursing interventions, 65.9% (*n* = 205) considered it to have equal importance compared to other nursing interventions; 19.6% (*n* = 61) considered it to have greater importance than other nursing interventions; 13.2% (*n* = 41) considered it to be able to replace other nursing interventions; and 0.6% (*n* = 2) considered it to have less importance than other nursing interventions. In addition, 0.6% (*n* = 2) considered it to not be important.

The analysis of the association between the importance attributed to HE practice revealed the existence of statistically significant differences with the “academic/professional qualifications” (χ^2^ = 89.008; df = 16; *p* = 0.009), those being the values with statistical significance at the level of nurses with only Master’s degrees. In other words, they give greater importance to the HE practice compared to other nursing interventions, as there is a higher than expected number of responses (2.0). Conversely, a lower than expected number of responses (−1.5) was observed in nurses who have a Master’s and Post-Graduation of Specialization. There was also a higher than expected number of responses in postgraduate nurses (8.8) when they attributed little importance to the HE practices (Table 1).

The analysis of the association between the importance attributed to HE practices and the “place of professional practice” revealed the existence of statistically significant differences (χ^2^ = 19.802; df = 4; *p* = 0.002), which were the effect on nurses who work in primary health care. In other words, from the analysis of the adjusted frequencies, values with statistical significance were found in both groups, although in reverse. There was an increase in the expected values in the responses referring to a greater appreciation of the HE practice in nurses who work in primary health care, whereas, in nurses who work in a pediatric hospital, there was a decrease. The data indicated, therefore, that there was a greater attribution of importance to HE practice compared to other nursing interventions in nurses who exercise their activity in primary health care (Table 2).

### 3.3. HE Evaluation to the Child/Young/Family

Regarding the assessment that more comprehensively reflects the HE contribution to the child/young and parents’ health portrayed by a scale—EAEPS (HE evaluation to the child/young and parents (EAEPS)—it was found that, as a whole, it was positively evaluated by nurses, since the percentage of agreement was higher than 70% on all items. The nurses also considered HE practice to essentially be about valuing the parents as a structure with functions and resources that affects the health and disease processes of the child/young, the adoption of healthy lifestyles and the identification of risk factors that harm the child/young/and parents’ health.

## 4. Discussion

According to the scientific literature, HE interventions have been shown to be useful [15], with nurses being recognized as the leaders of health promotion strategies by the World Health Organization (WHO) [16]. In this sense, it is essential to characterize the HE practices developed by nurses, namely to children/young and parents, in order to contribute to the improvement and the effectiveness of interventions.

In the present study, 77.5% (*n* = 241) of nurses perform HE interventions daily, which was expected once the criteria of the target population were that HE interventions were part of nurse’s professional activity. Although the literature points out nurse’s practices, in the context of child health, centered on the disease and developed in an individualized and fragmented way, especially when there is an overload of work and an excess of administrative activities [1], this result reinforces the idea that nurses are in an optimal position to identify problems and needs, to develop an early intervention and to promote the maintenance and/or improvement in the health status of children/young and parents [16]. In addition, recent studies indicate that the development of parenting increases maternal/paternal responsibility, which will naturally have a positive impact on child development [4]. In addition, Palmeira et al. [11] found a positive assessment of HE strategies by nursing, mainly for promoting new learning, training for conscious and adequate choices and providing greater awareness for self-care.

It should be noted that, in Portugal, the National Child and Youth Health Program focuses on issues related to child development, and nurses assume an important role in health promotion, prevention diseases and early intervention with children, teens and families at risk of psychosocial problems [16]. Consequently, the importance of the diagnosis of the population’s needs is understood, because the lack of planning HE practice can compromise the scrutiny on the distinctiveness of each individual, and the way they receive the guidelines and understand them [17].

In the present study, 65% (*n* = 202) of nurses admitted to preparing the HE practice according to children/young and parent’s needs, which reveals an adequate, targeted and effective response. Other studies corroborate this result. Kim and Sim [18] pointed out the importance of understanding the context and problems of individuals for the success of nurse’s interventions, and Heo and Lim [19] found that efficiency and the quality of nursing services is demonstrated when abilities are presented to perceive the individual’s problems.

The literature also points out the importance of welcoming and listening in order to achieve a close relationship, as this allows for the creation of bonds and, consequently, knowing the needs of each individual [20], and is the community singularity identification that allows for the promotion of a directed and quality HE practice, generating reflections and helping with autonomy [21].

As health-promoting attitudes and behaviors are important focuses in HE, healthy lifestyles promotion should be included in public health policies, namely those related to food, as it is a co-responsibility, i.e., it responds to personal, conscious and voluntary decisions shaped by parents and influenced by friends, teachers and the school environment [21]. In fact, the WHO report entitled “Health for the World’s Adolescents: A Second Chance in the Second Decade (H4WA)” points out that, globally, 80–90% of teens do not follow the norms of healthy eating behaviors [22]. Bearing in mind that, in the present study, healthy eating was considered the most important topic to be approached in HE (61.4%; *n* = 191), the nurse’s contribution to the training of good health practices is evident, as well as the contribution to the public policies efficiency. In addition, in an international study carried out in 2016 with 141 universities from 48 countries, the theme of healthy eating proved to be an area where a large investment had been made [23].

It is important to mention that, in the present study, the “Accidents prevention”, the “National Vaccination Plan” and the “Harmful behavior prevention” were also mentioned as being important topics. Thus, this reinforces the importance of the promotion of health-enhancing behaviors and the empowerment of the children/young and parents, allowing parents to make health decisions, which corroborates the studies by Buhr and Tannen [24] and Morrison et al. [25] that showed the relationship between children’s health and well-being and the health literacy demonstrated by parents. García et al. [26] also showed that the participants valued the families’ empowerment in the first place, followed by topics related to the prevention of accidents, diseases and health problems. Indeed, in order to promote health and well-being, it is essential that nurses, taking into account the binomial children/young and parents, regularly promote healthy lifestyles, which is also recommended in the health promotion model of Nola Pender, whose actions should promote well-being and self-updating [27].

Assuming that obstacles can prevent the good development of HE practices, it was important to know the elements that can facilitate or hinder the HE practice. A total of 74.6% of nurses considered the fear of a breach in confidentiality by children/young and parents as the element that most hinders the HE practices, expressing the importance attached to the limits of privacy/confidentiality. Indeed, ethical issues have been the subject of investigations [28]. The literature also points out the professional’s difficulties with adolescents, often justified by the lack of knowledge related to adolescence, besides gaps in training, reinforcing the importance of professional training and permanent education [20]. Barreto et al. [21] pointed out that the involvement promotion and approach of individuals with health professionals is the fundamental link to enable trust and respect in order to achieve the expected results.

Since the scientific evidence related to intervention programs in higher education show little relation to the requirements of the health promotion contexts and few programs with the criteria of good health promotion practices [29], it was concluded that it would be very useful to reinforce the importance of academic training in the context of the relationship with the children/young and parents. Other studies point out elements that hinder the HE practice. García et al. [26] found that there was a lack of time, reported by 71.4% of nurses. Einloft et al. [30] pointed out both the overload of nurses due to incomplete teams and bureaucratic management issues as the main justifications for the promotion and prevention issues. In the study by Ramos et al. [5], the work overload was pointed out, along with the lack of nurse autonomy in identifying the population needs and, above all, the lack of HE professional qualification. Souza et al. [1] also mentioned the lack of training, recognizing that theoretical–practical training is essential, and that the little use of protocols is also essential to guide the work of nurses.

Regarding which element most facilitate the HE practices, the present study revealed it to be the children’s/young’s and parents’ health priorities identification (89.7%; *n* = 279). This result highlights once again the importance attributed to the population needs diagnosis, which corroborates with the fact that the literature points out the importance of nurses, as health professionals with a broad role in which they identify problems, interrupt negative development trajectories and promote healthy behaviors and lifestyles [16]. Thus, once again, it is highlighted that a practice based on the needs identification produces a space of approximation between nurses, children, parents and the context in which they are inserted, establishing a health promotion practice guided by educational interactions [1].

In this line of thought, the academic training and qualification of nurses will be important, because educational practices are important for the development of competencies in health professionals [17]. On the other hand, the success of health promotion interventions strongly depends on the active individual’s participation, so the professional’s role must also be seen as facilitator and should be seen as active agents in the process [23].

In order to allow for a better HE practice evaluation, its importance was also questioned in comparison with other nursing interventions. The literature points out health promotion as a practice that increases knowledge, self-care and autonomy; that can influence social determinants of health; and can stimulate the individual to make decision and public policy planning [20]. In addition, according to the WHO [31], limited health literacy is associated with lower participation in the process of health promotion and disease prevention, so it is necessary to empower children, young and parents to decision making that contributes to the adoption of a healthier lifestyle. In Portugal, there was a high level of education, the younger had a high level of literacy, there was a positive correlation between health literacy and daily literacy practices and the privileged way to access health information was through health professionals [10].

In the present study, the majority of nurses (65.9%; *n* = 205) considered the HE practice to have equal importance to other nursing interventions.

This result can be understood by the fact that nurses do not see the effective results of their work and may feel discontented with current policies, due to the feeling of little recognition of the social value of nursing care, which, in turn, constitutes an important motivating tool for better care. In fact, studies show that nurses’ actions are largely concentrated in administrative and bureaucratic jobs, acquiring a less caregiver-type profile [1]. In addition, Barreto et al. [21] pointed out that HE is understood as a tool for transmitting information, which leads to a reductionist and positivist view of this practice by the professionals. Thus, there is the need to value the HE actions and to understand the importance of each professional category in this practice. Ramos et al. [5] evidenced the nurses’ dissatisfaction with the results achieved from the educational practices they develop, and, as a proposal to qualify the educational practices, they pointed out two educational dimensions: the training of nurses and the adoption of strategies that promote dialogue between professionals and population. The literature also points out the lack of time given to HE practice due to work overload, the scarcity of human and physical resources and a deficiency in the professional’s academic training, reducing the HP understanding toward prevention [20].

It should be noted that, in the present study, it was found that academic/professional qualifications (χ^2^ = 89.008; df = 16; *p* = 0.009) and the place of professional practice (χ^2^ = 19.802; df = 4; *p* = 0.002) influence the importance that nurses attribute to HE practice, i.e., nurses who have a Master’s degree and those who exercise their professional activity in primary health care are those who attach greater importance to HE practice compared to other nursing interventions. We can postulate the idea that academic qualifications enhance the deepening of the nursing theoretical practices, as well as professional activity, in primary health care. In turn, these are the fundamental pillars of all health systems in the world, and nurses are their foundations; therefore, they have the best position to influence the individual’s health status and well-being [32], as the nurses who work in hospital services focus more on curative interventions.

In view of this study’s results, it is highlighted that nurses develop a HE practice with the ability to influence the determinants of health and to promote the maintenance and/or improvement of the children/young and parents’ health status, which inherently leads to health gains and reinforces the effective contribution of HE performed by nurses to children/young and parents. These results also reinforce the nurse’s role as the main precursor of educational actions in health promotion, especially because, historically, they are responsible for producing comprehensive care and acting as educators in promoting health practice [33].

### Limitations

As a limitation of the study, the need for a larger group of participants can be pointed out, specifically the children/young, because the opinion of those who are the target of health care was important, as it brought evidence of recognition and a positive assessment of HE practices performed by nurses. It would be interesting in future research to obtain a different sample. On the other hand, as the present study was confined to a restricted geographic area, it would be beneficial for the improvement of nursing practice for the investigation to be consolidated through more comprehensive works in order to obtain generalized results at a national level.

## 5. Conclusions

The results of the present research revealed that nurses performs HE daily; plan the HE according to the identified needs of the target population; consider “healthy eating” as the most important topic to be approached in HE; consider that the most facilitating element in HE practice is the identification of the children/young and parents health priorities. It was found that the element that most hinders the HE practice is the fear of a breach in confidentiality by children/young and parents.

It is concluded, therefore, that, due to both the existence of taking advantage of the opportunities afforded by HE practice and the appreciation of the binomial children/young and parents for better needs identification, which produces more health conscious changes. Additionally the health-promoting attitudes and behaviors are important focuses in HE and are included in public health policies. Therefore, nurses showed the importance attributed to the current legal guidelines of the children/young and parent’s health, thus contributing to health gains. Consequently, it reinforces the nurse’s position as health educators.

This investigation also contributed to fueling the debate around HE practices in order to reinforce the role and the effective nurse’s contribution in HE practices to children/young and parents.

## Figures and Tables

**Table 1 children-09-00486-t001:** Results of the chi-square test by Monte Carlo simulation applied to the variables “academic/professional qualifications” and “importance of HE practices”. Trás-os-Montes and Alto Douro, Northern Portugal, Portugal, 2018 (*n* = 311).

		Importance of HE Practice						Chi-Square Test		
Academic/ Professional Qualifications		It Is Not Important	It Has Less Importance	It Has Equal Importance	It Has Greater Importance	Can Replace Others	Total	Value	df	*Monte Carlo Significance (2 Sided)*
Graduation	Count	1 ª	0 ª	116 ª	35 ª	24 ª	176	89.008 ª	16	0.009 ᵇ
	Expected count	1.1	1.1	116.0	34.5	23.2	176.0			
	Adjusted residues	−0.2	−1.6	0.0	0.1	0.3				
Post-Graduation of Specialization	Count	0 ª	1 ª	52 ª	17 ª	13 ª	83			
	Expected count	0.5	0.5	54.7	16.3	10.9	83.0			
	Adjusted residues	−0.9	0.7	−0.7	0.2	0.8				
Master	Count	0 ª	0 ª	7 ª	5 ª	0 ª	12			
	Expected count	0.1	0.1	7.9	2.4	1.6	12.0			
	Adjusted residues	−003	−0.3	−0.6	**2.0**	−1.4				
Master and Post-Graduation of Specialization	Count	0 ª^,^ᵇ	1ᵇ	29 ª^,^ᵇ	4 ª	4 ª^,^ᵇ	38			
	Expected count	0.2	0.2	25.0	7.5	5.0	38.0			
	Adjusted residues	−0.5	1.6	1.4	**−1.5**	−0.5				
Post-Graduation	Count	1 ª	0 ª^,^ᵇ	1 ᵇ	0 ᵇ	0 ᵇ	2			
	Expected count	0.0	0.0	1.3	0.4	0.3	2.0			
	Adjusted residues	**8.8**	−0.1	−0.5	−0.7	−0.6				
Total	Count	2	2	205	61	41	311			
	Expected count	2.0	200	205.0	61.0	41.0	311.0			
Likelihood Ratio								22.750	16	0.050 ᵇ
Fisher’s Exact Test								26.261		0.066 ᵇ
Linear Association								2.601 ^c^	1	0.106 ᵇ

ª 15 cells (60.0%) have expected count less than 5. The minimum expected count is 0.01. ᵇ Based on 10,000 sample tables with initial value 126,474,071. ^c^ The standardized statistic is −1.613.

**Table 2 children-09-00486-t002:** Results of the chi-square test by Monte Carlo simulation applied to the variables “place of professional practice” and “importance of HE practices”. Trás-os-Montes and Alto Douro, Northern Portugal, Portugal, 2018. (*n* = 311).

		Importance of HE Practice						Chi-Square Test		
Place of Professional Practice		It Is Not Important	It Has Less Importance	It Has Equal Importance	It Has Greater Importance	Can Replace Others	Total	Value	df	*Monte Carlo* *Significance (2 Sided)*
Primary Health Care	Count	2 ª^,^ᵇ	1 ᵇ	166 ᵇ	59 ª	41 ª	269	19.802 ª	4	0.002 ᵇ
	Expected count	1.7	1.7	177.3	52.8	35.5	269.0			
	Adjusted residues	0.6	−1.5	**−4.0**	**2.6**	2.7				
Hospital	Count	0 ª^,^ᵇ	1 ᵇ	39 ᵇ	2 ª	0 ª	42			
	Expected count	0.3	0.3	27.7	8.2	5.5	42.0			
	Adjusted residues	−0.6	1.5	**4.0**	**−2.6**	−2.7				
Total	Count	2	2	205	61	41	311			
	Expected count	2.0	2.0	205.0	61.0	41.0	311.0			
Likelihood Ratio								26.359	4	0.000 ᵇ
Fisher’s Exact Test								22.629		0.000 ᵇ
Linear Association								14.966 ^c^	1	0.000 ᵇ

ª Four cells (40.0%) have expected count less than 5. The minimum expected count is 0.27. ᵇ Based on 10,000 sample tables with initial value 1,507,486,128. ^c^ The standardized statistic is −3.869.

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
