# Peer review of "Nurse’s Evaluation on Health Education in Portuguese Pediatric Hospitals and Primary Care for Children/Young and Parents"

_children, 2022, doi:10.3390/children9040486_

Round 1

Reviewer 1 Report

A very interesting paper due to the topic it addresses, which highlights the important role of nursing in health education in childhood and adolescence.

Introduction. To situate health education in childhood and adolescence, the figure of the school nurse cannot be forgotten and the good results they have in improving the health of children and youth in the countries where this figure has been developed.

Methodology and Results. Good use of quantitative methodology. The results are clear and well presented.

Discussion and Conclusions. Very complete, including the difficulties of nursing and future challenges in health education in childhood and adolescence and its relevant role in this field.

When talking about the role of nursing in health education, the experiences of school nurses and the vindication of this figure are missed, as well as the network of “Schools for health in Europe” and their strategy to improve the health of children and teenagers.

Author Response

The reviewer's suggestions have been accepted and included in the revised version of the paper.

Reviewer 2 Report

Dear Authors,

Your paper deals with a very important topic in the context of health equity of and health promotion for children. In the following, please find my comments and recommendations.

Title: The title of your paper would be more suitable for a review paper. For your paper relates to Portugal, I recommend to include this information in the title.

Abstract: Please, add the background of the study, shorten the summary of results and include a discussion paragraph.

Keywords: keywords should not repeat words/terms already used in the title. Please, replace.

Introduction: At the moment the introduction is a string of excerpted literature and lacks sufficient information on the relevance of health education evaluation related to children, youth and parents in Portugal.

Please, provide more information on the health status of children and youth and the parents’ health literacy in Portugal. Furthermore, please, provide information on the knowledge gap and point out the urgency of dealing with the topic.

Please, shift relevant information presented in the Discussion chapter to this chapter.

General remark: Please, add chapter number.

Methods: Please, put your research in context (background of your study). Information above all on methodological considerations related to the selection of units (hospitals), information on content and structure of the questionnaire as well as the data collection process (written survey!) is missing.

General remark: Please, add chapter number.

Results: This section lacks structure and is difficult to read, because relevant information on the content of the (written?) survey. Please, insert sub-headings. Provide more (concise) tables on statistical results! Shift large tables to the annex.

Discussion: This section contains information which are not appropriate for this chapter. For instance, general information on health education interventions, statistical results as well as policy recommendations. Moreover, this section relates to important policy documents (cf. line 213 to line 215) that are not mentioned before; please, insert sub-headings in order to increase readability.

Limitations: please, provide more methodology-related considerations.

General remark: Please, add chapter number.

Conclusion: is not sufficiently supported by the empirical results and should be more accurate.

General remark: Please, add chapter number.

Kind regards

Author Response

(The authors gave the same response as above.)
